# What Turns a Product into a Traditional One?

**DOI:** 10.3390/foods10061284

**Published:** 2021-06-04

**Authors:** Sergio Erick García-Barrón, Luis Guerrero, Ariel Vázquez-Elorza, Oxana Lazo

**Affiliations:** 1CIATEJ Research and Assistance Center in Technology and Design of the State of Jalisco A. C. Av. De los Normalistas #800, Colinas de La Normal, Guadalajara, Jalisco 44270, Mexico; segarcia@ciatej.mx (S.E.G.-B.); avazquez@ciatej.mx (A.V.-E.); 2IRTA—Food Quality and Tecnhology, Finca Camps i Armet s/n, Monells, 17121 Girona, Spain; lluis.guerrero@irta.cat; 3Research Center for Applied Biotechnology, CIBA Instituto Politécnico Nacional, Carretera Estatal Santa Inés Tecuexcomac–Tepetitla, km. 1.5, Tepetitla de Lardizábal, Tlaxcala 90700, Mexico

**Keywords:** traditional dimensions, habit, product involvement, objective knowledge, subjective knowledge, consumer behavior, alcoholic beverages

## Abstract

Consumer interest in traditional food products (TFPs) has increased in recent decades. The concept of TFPs is made up of seven dimensions. However, it is not yet clear what the contributions of these dimensions are to the perception of the traditional image of a specific product. In addition, the effects of constructs such as habit, product involvement and objective and subjective knowledge on the traditional character of a product have not been explored either. The aims of this work were to evaluate the influence of the dimensions of the traditional food concept on the perception of mezcal and tequila and to understand consumer’s perception of the traditional character of the beverages through their segmentation characteristics. Eight hundred consumers were surveyed in four Mexican cities. A questionnaire was designed to assess the constructs, TFPs’ dimensions, sociodemographic information and consumption patterns. Results showed that the dimensions of the traditional concept allowed a better understanding of the traditional character of the product, as well as their individual relevance showing that frequent consumption is not always linked to the traditional character of a product. Three clusters were obtained for both products based on the assessed dimensions of the traditional concept. The presence of the segments showed variations in the contribution of the different dimensions to the concept of “traditional”. Geographic location, special dates and sensory dimensions are determinant in the traditional perception of both beverages, which is useful to design effective strategies to promote rational and responsible consumption.

## 1. Introduction

Nowadays, there is increasing demand for traditional food products [1]. Therefore, a positive valorization of traditional food has been made by consumers [2], especially when products are based on technical, economic, social, patrimonial, cultural and environmental characteristics [3].

Different assessments of traditional food products have been made over the years. Guerrero et al. [4] evaluated a cross-cultural conceptualization between traditional and innovation concepts in six European countries. Đorđević and Buchtova [5] studied ethnocentrism as a factor that influences the acceptability of traditional and nontraditional products. Roselli et al. [6] focused on willingness to buy traditional olive oil, and Silvestri et al. [7] studied the perception of quality of traditional Parmigiano Reggiano cheese. However, none of these studies addressed the rationale behind the traditional character of a product. Consequently, it seems important to analyze what makes a product traditional from a consumer’s perspective. In this vein, Guerrero et al. [8] defined a “traditional food product” from a consumer point-of-view. This definition comprises seven relevant dimensions to be taken into account when assessing the traditional character of a product. Nevertheless, the application of these dimensions in specific products should be tested to be validated.

In addition, and in order to better comprehend what makes a product traditional, it is also crucial to understand consumers’ attitudes and opinions towards it. Since there are different types of consumers, it is essential to know their specific characteristics and in which domains they interact [9]. To achieve this, segmentation is necessary to explain their behavior [10]. When segmenting, it is important to take in to account, not only sociodemographic variables, but also to include consumers´ experience with the product, consumption patterns and psychosocial characteristics such as habit [11], product involvement [12] and knowledge [13], which could help form a better understanding of their opinions towards traditional products.

It seems that most consumers associate a traditional food with habit [8]. Habit has been noticed to be a strong element in food-related behavior [8]. Consumers are individuals with inner habits; they tend to replicate conducts as an automatic response without considering further consequences in their actions. Habits are learned by creating objectives in life or by making an effort to break undesired comportment [14]. Therefore, modification of consumer habits could be an indication of change in the perception of a traditional product [15].

Product involvement has been a topic of interest in consumer research since it can influence consumer behavior [16]. Involvement can be defined as the level of perceived personal importance and the interest or relevance evoked by a stimulus, which are linked by consumers’ ability to endure situation-specific goals [17]. From a behavioral point of view, involvement is a multidimensional construct. In the case of some alcoholic beverages such as wine, the effect of involvement on consumers’ behavior is generally of a durable nature, depending on the consumption situation [18]. Nevertheless, and even though the role of product involvement in different traditional products has been previously studied [19], its influence on the perception on the concept of “traditional” has not been addressed.

According to Ellis and Caruana [20], consumers’ knowledge of a product significantly affects their purchase behavior, when it is consumed and how it is experienced. Knowing the history behind a product can be a key element driving the perception of its traditional character. For example, knowing the product is linked to a cultural heritage or that it contributes to the development and sustainability of rural areas may improve their view of traditional food [8]. Consumer knowledge is a concept formed mainly by two factors: expertise and familiarity [21]. The expertise is made up of two constructs, namely subjective and objective knowledge. Subjective knowledge represents consumer´s personal perceptions of what and how much he or she knows about a product. Objective knowledge on the other hand, characterizes the stored information and its organization in the memory, which refers to what the consumer actually knows about the product [21]. Familiarity may arise from personal expertise with specific foods accumulated through purchasing and consuming these products [22]. Although the impact of these constructs on consumer behavior has been demonstrated previously, their influence on the perception of a traditional food has not been addressed in depth.

Therefore, in this work mezcal and tequila (two alcoholic beverages with different history and technological processes associated with them) were selected as cases of study since they are considered part of Mexican culture and are associated with high social and economic impact [23]. Mezcal is a beverage that is normally elaborated through a completely artisanal process and with different varieties of agave depending on the place of origin, which gives it sensory characteristics according to the region where it is made [24]. On the other hand, due to the increase in national and international consumption, tequila is a beverage with a technical production process in which only blue agave can be used [25]. Additionally, the consumption of mezcal and tequila has grown significantly in recent years [23,26], especially since they have Protected Designation of Origin and belong to specific sociocultural regions. Moreover, to our knowledge the traditional character of distilled alcoholic beverages has not yet been studied, the results of which could be of interest to the alcoholic beverages industry.

Even though these two products are widely perceived and accepted as traditional products, their traditional nature has not been assessed from a consumer’s point of view, and more specifically through a psychographic approach. Thus, consumer segmentation could be a tool to address marketing strategies based on each group of consumers’ characteristics.

Therefore the aims of this work were: a) to evaluate the influence of the dimensions of traditional food concept provided by Guerrero et al. [8] on the perception of two Mexican alcoholic beverages and b) to understand consumers’ perception of the traditional character of the beverages through their segmentation characteristics.

## 2. Materials and Methods

### 2.1. Participants

A total of 800 individuals from four different Mexican cities participated in this study (Table 1). The included cities were Mexico City (CDMX), for being the largest city of consumption of mezcal [27] and tequila [28] in the country; Guadalajara (GDL) (occidental zone of Mexico), wherein mezcal consumption is rather incipient and is the region of origin of tequila; Oaxaca (OAX) (southeast area of the country) as the leader of mezcal production in the country; and Puebla (PUE) (east zone), as a mezcal producer region (since it has recently received a protected designation of origin title for the beverage) and also a growing consumption market. In each city, 200 participants were surveyed face-to-face, 100 mezcal consumers and 100 tequila consumers. Criteria for selecting participants were their interest in participating in the study, being over 18 years old, to be resident (more than 10 years) in the corresponding selected city and being a mezcal or tequila consumer on at least a monthly basis. A convenience sampling was performed, which is a frequently used methodology in social and marketing research wherein participants are selected by accessibility and proximity to the interviewer [29]. All participants were surveyed in places where mezcal or tequila is usually consumed (bars, canteens and local fairs and exhibitions).

### 2.2. Survey Design and Implementation

A slightly different questionnaire was designed for both mezcal and tequila consumers. Each questionnaire had two sections (Table 2). The first section was oriented to evaluating four constructs: habit [11]; product involvement [12]; objective knowledge (own elaboration) and subjective knowledge [13]. In addition, the “Overall perceived traditionality” of mezcal or tequila was also assessed. All constructs were evaluated in a seven-point Likert scale, from 1 (“totally disagree”) to 7 (“totally agree”) with the sole exception of objective knowledge and the “Overall perceived traditionality”.

The objective knowledge was evaluated by means of a 15 multiple-choice questionnaire formulated by professionals of the mezcal and tequila industries in collaboration with the authors of this work (Appendix A). The “Overall perceived traditionality” of the two products was assessed on a nine-point scale (1 “not traditional at all to 9 ”very traditional”), with which participants indicated how traditional they perceived the product (mezcal or tequila) to be. In addition, product traditionality was also assessed according to the definition proposed by Guerrero et al. [8], which contains seven different dimensions. However, due the nature of the products analyzed in this study, two of these dimensions “Product consumption is something I inherited from my parents” and “The product is elaborated according to a gastronomic heritage” were eliminated. Neither mezcal nor tequila are normally taught to be consumed from an individual´s ancestors, so this dimension was not included in the questionnaire. In the case of gastronomic heritage, and according to Rojas-Rivas et al. [30], Mexican consumers relate “gastronomy” with different aspects, among them recipes to prepare the product. For this reason, and in order to avoid misinterpretations, we did not include this dimension of the traditional concept, since most consumers do not prepare these beverages by themselves. The selected dimensions of the traditional concept were also assessed by means of a seven-point Likert scale (from 1 (“totally disagree”) to 7 (“totally agree”)).

The second part of the questionnaire was focused on sociodemographic information from the participants (age, gender, place of residence, etc.), as well as mezcal and tequila consumption habits.

### 2.3. Data Analysis

Reliability and internal consistency for habit, product involvement and subjective knowledge constructs were evaluated through Cronbach’s alpha coefficient. In addition, and to check the unidimensional character of the constructs, a factor analysis (principal components extraction method) was performed as well for each one of them. These analyses were performed for each alcoholic drink (mezcal and tequila) separately.

Once unidimensionality and internal reliability were confirmed (values greater than 0.6 for both factor loadings in the first dimension and Cronbach’s alpha and in both drinks), mean values of the items included in the constructs were calculated, and the new composite variable was retained. For the objective knowledge, the total number of correct answers was counted (between 0 and 15). Regarding the different dimensions of the traditional concept, these analyses were not performed. According to Guerrero et al. [8], although these dimensions tend to be internally consistent, people do not necessarily have to agree with all of them. For this reason, these dimensions have not been aggregated.

A two-way analysis of variance was performed over the different constructs and for each of the five dimensions of the traditional concept assessed. The model included the location (different cities studied) and the participant as fixed and random effects, respectively. When significant differences were observed (*p* < 0.05), a post hoc test for multiple comparisons of the mean values was performed through Tukey’s HSD test.

To detect the presence of different segments of individuals with similar response patterns based on the scored dimensions of traditional concepts, a hierarchical cluster analysis (Ward method and Euclidian distance) was performed for each alcoholic drink. The number of segments to retain was decided based on the obtained dendrogram, considering the homogeneity intra- and inter-segments [31] and the principle of parsimony. The selected segments were validated by means of a discriminant analysis to assess the percentage of individuals correctly classified in their respective cluster. Afterwards, a two-way ANOVA was performed over the scores of the five dimensions of the traditional concept, including the cluster and the consumer as fixed and random effects, respectively. To determine significant differences between segments, a post hoc Tukey´s test (*p* < 0.05) was performed. This same statistical linear model was used to characterize the segments based on the evaluated constructs (habit, product involvement and objective and subjective knowledge). In addition, a Chi square test of the K proportion analysis using the Marascuilo procedure was performed to identify the existence of differences between segments (*p* < 0.05) depending on the sociodemographic characteristics of the participants and their corresponding consumption habits. All the analyses were carried out by means of XLSTAT 2018 software (Addinsoft, Paris, France).

## 3. Results

### 3.1. Sociodemographic Characteristics of Participants

Table 1 shows the sociodemographic characteristics of mezcal and tequila´s consumers. In general, there were a higher number of men participants than women. Regarding the age of the participants, in most cases more than 60% of them were between 25 and 44 years old. Concerning consumers’ experience with the products, 29.25% of the mezcal consumers (*n* = 117) have been drinking the beverage for over seven years, and 11.75% (*n* = 47) for less than a year. On the other hand, when it comes to tequila, 49.5% (*n* = 198) have been drinking the product for more than seven years and only 4.5% (*n* = 18) for less than a year. In general, the consumption of mezcal seems to be more frequent than that of tequila. Thus, 32.5% of the participants stated that they had a mezcal consumption of at least once a week compared to the 21.3% observed for tequila.

### 3.2. Unidmensional Structure and Reliability of the Different Constructs Multi-Item

The internal reliability of the assessed constructs (Cronbach’s alpha values) for mezcal was 0.7 for habit, 0.9 for product involvement and 0.8 for subjective knowledge. For tequila, the obtained values were 0.8 for habit, 0.8 for product involvement and 0.6 for subjective knowledge. According to George and Mallery [32], values from 0.5 to 0.6, even being rather low, can also be accepted. The corresponding factorial loadings for mezcal were >0.65 for habit, >0.83 for product involvement and >0.68 for subjective knowledge. For tequila, these values were >0.62, >0.79 and >0.71 for habit, product involvement, and subjective knowledge, respectively.

It is also worth mentioning that the subjective knowledge initially contained 6 items. However, the item “I rarely find a brand of Mezcal (Tequila) that I have not heard about” showed a very low factor loadings in the first dimension of the factor analysis in both beverages (0.179 for Mezcal and 0.001 for Tequila). Thus, this item was eliminated from the construct for both beverages.

### 3.3. Overall Perceived Traditionality and Constructs Per City and Product

Table 3 shows the results of the “Overall perceived traditionality” and the constructs evaluated in each tested city. Even though statistical differences were observed between cities for both beverages (*p* < 0.05), consumers perceived both mezcal and tequila as highly traditional beverages. No significant differences were observed for the habit construct of mezcal consumers among the cities. In the case of tequila, consumers from GDL exhibited the highest mean value, although it was only significantly different from Puebla consumers.

Regarding product involvement, significant differences were observed for the consumers from OAX for mezcal and from those from GDL for tequila. Concerning the objective knowledge of mezcal, participants from CDMX, GDL and PUE were characterized for knowing less about the beverage than participants from OAX. Likewise, regarding subjective knowledge, participants from OAX had the highest scores as opposed to consumers from GDL, who were the ones who thought they knew less about mezcal. On the other hand, when it comes to tequila, GDL consumers can be defined as the ones with the highest objective knowledge of the beverage, followed by the CDMX and OAX consumers, and the participants from PUE being the ones with the lowest knowledge of the beverage. Regarding subjective knowledge, participants from GDL also showed higher scores than the consumers from the other three cities.

### 3.4. Differences in Dimensions of the Concept “Traditional Food Product”

Based on the traditional food product definition from Guerrero et al. [8], the selected five dimensions that build the concept of “traditional Food” differed among participants from the different cities for both beverages (Table 3). Mezcal consumers from OAX tended to show higher mean values for the five dimensions of the concept of “traditional” and, in the case of tequila, consumers from GDL showed the same pattern except for the “low process” dimension. In the case of the “regular consumption” dimension, even though significant differences were observed between cities for both beverages, all participants shared the idea that both mezcal and tequila are not products for regular consumption. In any case, mean values in this dimension tend to be higher for mezcal than for tequila.

### 3.5. Segment Identification and Characterization

#### 3.5.1. Mezcal Segments

The hierarchical cluster analysis based on the five scored dimensions of the traditional concept showed three different mezcal consumer segments. Based on the confusion matrix obtained from the discriminant analysis, 92.0% of consumers were correctly classified in their respective cluster. Table 4 shows sociodemographic characteristics, consumption-related features, mean construct values and mean values for dimensions of the traditional concept for each segment.

The sociodemographic variables that showed differences between segments were city of origin (CDMX and GDL) and age range (45 to 54 years old). The consumption related variables also differed between clusters for experience with the product (three to five years and over seven years) and frequency of consumption (every day, every third day and once a month). Significant differences between segments for all constructs (*p* < 0.05) were observed, with the sole exception of the “Overall perceived traditionality”, which implies that all consumers perceived the beverage as having similarly “traditional” character. Regarding the dimensions of the “traditional” concept, “regular consumption”, “geographic location” and “low process” exhibited significant differences between segments.

Cluster One was characterized for being the only one who disagrees that mezcal is manufactured through low-tech processing conditions, and for this reason, it was named “Unfamiliar with the process” (35.25% of the participants). This group did not differ significantly from any of the other clusters regarding sociodemographic variables. It linked mezcal to a certain geographic location and mostly showed monthly consumption. In addition, this group (along with cluster three) also presented lower levels of objective and subjective knowledge.

Cluster Two was labeled as “Connoisseurs” (19.5% of the participants). This group was the smallest one and had the highest percentage of consumers from CDMX. This segment also had the largest percentage of participants (compared with the other two clusters) from 45 to 54 years old, with more than seven years of experience with the product and with the highest frequency of consumption (every day and every third day). This agrees with the highest habit mean value observed and the strongest idea that mezcal is a product of regular consumption. Consumers from this segment were also more involved and knew more about the beverage than the other two segments.

Cluster Three was tagged as “Uninvolved” (42.25% of participants). This group had higher percentage of participants with three to five years of experience with the product than the other two clusters. They also had the lowest levels of product involvement and as happened in segment one, they had a monthly consumption, which could explain the reason for not linking mezcal to a regular consumption. This group believed that mezcal is manufactured in low-tech conditions (showing the highest mean value in this dimension), which is the main differentiating factor from segment one.

#### 3.5.2. Tequila Segments

Three clusters of participants were obtained for tequila from the dimensions of the “traditional concept” (Table 5). Based on the confusion matrix, 93.3% of tequila consumers were correctly classified in the three clusters retained for this beverage. Place of origin, age (18–24 years old), experience with the product (one to three years and over seven years) and frequency of consumption (once a week and monthly consumption) showed significant differences between segments. The scores on the dimensions that are part of the “traditional concept” showed significant differences between segments (*p* < 0.05) in all cases. “Overall perceived traditionality” of the product was clear for all three segments (mean values higher than 8).

Segment One was named “Occasional consumers” (41.75% of the participants). This group presented the highest percentage of young adults (18–24 years old) among the clusters. This segment had a high percentage of participants with 1–3 years of experience compared to segments two and three. Their consumption of tequila is mainly monthly, and they did not agree that tequila is manufactured under low-tech conditions. This group presented the lowest level of habit and product involvement with a medium level of objective knowledge. However, they believed that tequila is linked to festivities and special dates, as well as having particular sensory characteristics.

Segment Two was defined as “Experienced” (46.5% of the participants). This group has a higher number of consumers who have been consuming tequila for more than seven years, and they believe that tequila is manufactured with a low-tech process. In addition, they had the highest scores on the perceived “Overall perceived traditionality” and presented the highest level of objective knowledge. Participants in this segment believed that tequila is not for regular consumption. As in segment one, in this group, tequila consumption is linked to special occasions or festivities and also that it is perceived has having distinctive sensory characteristics.

Segment Three was named “Uninformed” (11.75% of the participants). This was the smallest of all segments, having mostly OAX consumers. This segment also showed a high percentage of participants with over seven years of experience consuming the product like segment 2 and exhibited mainly a weekly consumption. In addition, this group also thought tequila is a beverage of regular consumption, and their habit and product involvement values were higher than the other two segments. However, this cluster had the lowest objective knowledge, which could explain their thoughts on tequila not being strongly linked to a geographic location. Regarding the dimensions “Sensory” and “Special dates”, this group showed lower score values than the other two segments. Nevertheless, this segment still agreed on the relevance of these two dimensions.

## 4. Discussion

This work assessed the dimensions that define a traditional product using the definition provided by Guerrero et al. [8]. Additionally, the role of habit, involvement and subjective and objective knowledge on consumers’ perception of the traditional character were also determined for two Mexican traditional beverages, namely mezcal and tequila, in four Mexican cities.

Consumers’ insight on what a traditional product is may vary according to the context in which they consume it. According to García-Barrón et al. [23], in the mid-90s Mezcal used to be a beverage drunk by low-income consumers associated with poor quality, whereas tequila was positively appreciated, increasing its economic value in the industry. It has been just in the past decade that mezcal’s image has improved in the eyes of consumers, thus explaining the drinking history of consumers with these two beverages. These authors also observed that consumers more frequently associate tequila with “party” or special occasions and mezcal with food or daily use, which totally agrees with the self-reported consumption behavior observed in our study. These associations could help to better understand the frequency in which each one of these beverages is normally consumed.

### 4.1. Comparison between Cities

The observed differences in the constructs and dimensions scores for both beverages show that, even within the same country, noticeable variations are found among consumers of the same product belonging to different regions and subcultures. As stated by Sobal [33], several cultural subgroups can be identified within each culture or country. It is relevant to mention that subcultures are important in complex societies that include different cultural groups, such as in Mexico. Subcultural comparisons should be performed between these groups to better understand collective behavior. It is important to mention that these differences are also relevant when dealing with highly traditional products, as observed in our study.

In most cases, consumers whose place of residence was the same where the beverages were traditionally produced showed the highest scores in the constructs and dimensions evaluated, which could be related to the frequency of exposure to the product [34]. Constant interaction with the product contributes to familiarity formation, which represents the accumulation of experiences related with the product. Familiarity also plays a preconditional role for knowledge [35], which might explain the reason for the highest scores for objective and subjective knowledge for participants from OAX for mezcal and GDL for tequila. In this vein, the relationship between knowledge and involvement could also be explained. According to Aurifeille et al. [36], the more involved consumers are, the higher their tendency to know more about the targeted product, which is consistent with the observed results for participants from OAX with mezcal (the leader of mezcal production in the country) and GDL (the region of origin of tequila) with tequila. Regarding those consumers from cities where these beverages are not produced, they did not seem have the same level of knowledge as ones from the cities where the products are manufactured. The observed relationship between these variables (product involvement, habit and both types of knowledge) opens multiple possibilities to explore how to promote the consumption of the two beverages outside their main production area.

### 4.2. Mezcal Segments

The presence of different consumer segments shows that the dimensions of the “traditional” concept are valued differently. In that sense, the belief that mezcal is not manufactured through low-tech conditions by consumers in segment one could be related to the low level of objective and subjective knowledge that they presented (along with segment three). Consumers of segment two on the other hand, did think of mezcal as an artisanal product (low-tech conditions). This segment has higher levels of objective knowledge on the product and thus, could have a tendency to seek information about it, which probably makes them aware of the production process of the product. Therefore, providing information to segments one and three of consumers about the artisanal character of the product and the impact that the production process has on the final properties could improve mezcal’s perception and the willingness to consume it. Accordingly, Caporale and Monteleone [37] pointed out that information about the alcoholic beverages process can influence consumers’ expectations and their further consumption. In addition, segment two had the consumers with higher levels of product involvement. According to Charters and Pettigrew [38], involvement can be considered a suitable indicator of knowing a product, as observed in the present study. In that sense, Aurifeille et al. [36] pointed out that the more involved consumers are, the higher their tendency to know more about the targeted product. In addition, Cowley, [39] suggested that a “connoisseur” possesses knowledge based on a broad personal experience, familiarity with the product and objective knowledge, which influences behavior towards the product [20]. This might suggest that consumers who have been consuming the product for a longer time (tending to be adults over 25 years old) are those who tend to know more about the product, since they have been drinking it for a longer time, as was observed for this segment of consumers. Finally, considering that segment was the largest segment, producers should not underestimate this group of consumers and should try to find the best way to involve them with the product. In this vein, Aurifeille et al. [36] suggest that that, in the case of consumers with low levels of involvement, strategies should be generated to improve the consumption experience.

According to the behavior observed in consumers from segment 2, subjects with a higher level of knowledge, involvement, frequency of consumption and experience with the product have a tendency to perceive mezcal as a more traditional product.

### 4.3. Tequila Segments

For tequila, differences and similarities were observed in the values of the dimensions of the traditional concept. It was observed that occasional consumers (segment one) had a higher number of young participants (18–24 years compared with the other two segments), with low levels of habit and product involvement. This might be related to the possibility that young consumers are just beginning to introduce themselves to alcohol consumption, mainly seeking fun and enjoyment while consuming these beverages without any further involvement with the product itself [40]. In the case of consumers of segment two and despite their higher level of objective knowledge, they did not think tequila was for regular consumption. This result could be related to what was observed by García-Barrón et al. [23], who found that the representation of the concept of “tequila” was linked to “parties”, which not necessarily need to be frequent, such as: “Independence Day” or “Town festivities” [41]. This association agrees with the obtained results, thus agreeing with the belief of consumers from cluster one and two on the presence of this beverage for special occasions or festivities, in which tequila has an important role. According to Crawford, [42], among the different reasons for people consuming alcohol are the “social” motives, which refer to social obligations (“being social is right”, “people that I know drink”) and celebrations (“celebrating a special occasion”), as could be the case of segment one and two.

Regarding segment three, it is worth noting that more than 60% of the participants came from Oaxaca. In this region, there is more exposure to mezcal than to tequila since this city is part of the leading state in the production of mezcal [27]. Therefore, tequila is not a local product for this segment, and thus, participants might see it as a more distant product that does not form part of their own traditions or habits. Consequently, the beverage consumption in this segment could be influenced by other factors, such as hedonic (liking), economic, (accessible price), advertisement and/or accessibility.

Taking into account the differences and similarities of these consumer segments, the traditional product image of tequila tends to be positively influenced mainly by the objective knowledge of the consumers, as observed in segment two.

### 4.4. The Dimensions of Traditional Concept

Despite the perceived “Overall perceived traditionality” of these two products, results show that not all segments think these are frequently consumed beverages. Even though regular consumption is an important dimension in food traditionality [8], it does not seem to affect the traditional perception of these two beverages by all the participants. This could be related to the high alcoholic concentration of these products, social, norms, values, attitudes and the risk of excessive alcohol consumption [43]. Mitchel and Graetorex [44] pointed out that buying and consuming alcoholic beverages implies financial, functional, physical and social risks, all of them probably affecting the observed results in our study when dealing with this dimension.

Even though significant differences between the segments of each beverage were observed for the “Geographic location” dimension, in all cases products were linked with a place of origin, especially in the case of mezcal. Both products have a Protected Designation of Origin [45,46], which could explain the high mean values observed for this dimension in the present study. The importance of this dimension relies on the premise that food origin (geographic location) can affect food choice and consumption, especially when it comes to traditional food [7,47]. This is also related to the idea of the origin concept (location) being a quality attribute, aside from the symbolic and emotional meaning for the consumers [48]. The bond between the product and the location might suggest that a product is manufactured from local raw materials that provide sensory attributes typical from the region where it is manufactured. These attributes make the product distinctive and even more attached to the place of origin [2,8].

The dimension “Low process” showed a stronger belief that mezcal has a low-tech elaboration process than the one that is used for tequila. Mezcal is normally manufactured in an artisanal manner, whereas tequila is normally made following a high technological process [49]. The manufacturing conditions of a product are determinant for its image as a traditional product. To be considered “traditional”, it is necessary for the product to have an manufacturing process with specific raw materials linked to a place of origin. Thus, high-tech processing would risk losing the traditional image of the product [50]. The lower consensus level between tequila´s consumers could be related to growing demand in both national and international markets causing mass production that may influence its cultural and traditional value [51]. The above could explain the higher scores of “Overall perceived traditionality” observed in mezcal consumers compared to tequila consumers.

Both mezcal and tequila consumers believe that these beverages have distinctive sensory characteristics, thus confirming the overall importance of the “Sensory” dimension in a traditional product. In fact, for mezcal, this dimension showed one of the highest mean-value scores. There are some studies that pointed out that consumers can distinguish and appreciate some specific sensory properties of these two alcoholic drinks [52,53]. According to Villanueva-Rodriguez and Escalona-Buendia, [49], consumers are expecting smoke notes in mezcal because of the traditional cooking of agave in ground holes with burning wood and heated stones that produce furans and smoky volatiles that are retained by the agave. On the other hand, consumers would appreciate cooked agave and floral sensory notes in tequila. The sensory characteristics of a product are considered a key quality factor for its acceptability or rejection. They represent a simple approach in identifying the authenticity of a food product, especially when it comes to the traditional character [8,54].

In the case of the dimension “Special dates”, participants associated both beverages with specific celebrations and/or seasonal occasions. However, tequila seems to be more associated with celebrations than mezcal. According to Gómez-Cuevas et al. [55], tequila has a celebratory value, which links it to special or social moments that do not necessarily take place daily. This relationship reflects that the consumption of both beverages, especially tequila, is given in a social context, as it has been observed in other alcoholic beverages from other cultures [43]. Alcoholic beverages have a strong sociocultural role that may impact the consumption context [56]. According to Douglas [57], alcohol consumption reflects an individual or group identity that describes consumers’ self-perception through those who consume it and the way they consume it. Alcohol consumption has a social meaning in many different cultures, thus being used when celebrating a special event [56,58]. There are different factors that may influence this relation, such as the information passed from one generation to another [8] and the promotional campaigns oriented to enhance the perception of representative products of Mexican culture, such as mezcal and tequila.

Overall, the results in this work suggest that the dimensions of sensory, special dates and geographic location are crucial for mezcal and tequila perception as traditional beverages. However, the low-process dimension is only important as a part of the traditional image for mezcal. Finally, the regular consumption dimension did not appear to be important for either Mezcal or Tequila when considering the traditional character of these two products.

The results of this work can be useful to design and implement more effective strategies and policies aimed at promoting more rational and responsible consumption of these two beverages while keeping their traditional meaning and values.

## 5. Conclusions

The dimensions proposed by Guerrero et al. [8] allowed us to assess and better understand the traditional character of mezcal and tequila. Participants from the geographical locations where these two products had higher familiarity and rooting scored the highest values in most of the assessed constructs and dimensions. This suggests that the consumption of these traditional drinks is part of their cultural identity.

According to the consumers from different segments, both mezcal and tequila were considered highly traditional beverages. However, it is worth mentioning that the traditional image is not always related to frequent consumption (as suggested in previous studies), especially when it comes to high-level alcoholic beverages.

The presence of these segments shows that there are variations in the dimensions contributing to the formation of the “traditional concept”. In addition, the tested constructs proved to have an influence on consumers´ perception of a traditional product. In the case of mezcal, product involvement and knowledge tended to be significant in the segments with higher values of traditionality, experience with the product and higher frequency of consumption. In the case of tequila, objective knowledge seemed to be decisive in the appreciation of the traditional image of the product.

The aforementioned results suggest that consumers consider mezcal and tequila to be a complex mixture of different dimensions, wherein sociodemographic characteristics, consumption habits, behavioral aspects and the nature of the product play a remarkable role. The results of this work can be useful to design and implement more effective strategies and policies aimed at promoting more rational and responsible consumption of these two beverages. These strategies could be based on highlighting the sensory characteristics of each drink derived from its traditional way of manufacture, how the drinks are linked to a graphic region and their relationship with special events. This should be applied to highlight the key elements that drive the traditional perception of a product and thus, help to keep its traditional image in market segments that are unfamiliar and/or possess little knowledge of these two traditional beverages.

## Figures and Tables

**Table 1 foods-10-01284-t001:** Sociodemographic characteristics (number of individuals) and consumption habits of the participants in the survey (*n* = 800).

	Mezcal Consumers				Tequila Consumers			
Variables and Categories	CDMX (*n* = 100)	GDL (*n* = 100)	OAX (*n* = 100)	PUE (*n* = 100)	CDMX (*n* = 100)	GDL (*n* = 100)	OAX (*n* = 100)	PUE (*n* = 100)
Gender								
Men	60	63	59	59	51	64	57	52
Women	40	37	41	41	49	36	43	48
Age								
18–24 years	10	16	24	12	21	15	34	10
25–34 years	39	46	40	37	42	43	44	31
35–44 years	23	29	27	34	18	21	15	36
45–54 years	20	7	7	14	10	12	6	13
55 or more	8	2	2	3	9	9	1	10
Experience with the product								
Less than a year	5	16	10	16	1	3	10	4
1 to 3 years	17	19	21	26	8	7	24	5
3 to 5 years	18	27	15	21	20	10	25	14
5 to 7 years	17	20	16	19	18	14	25	14
More than 7 years	43	18	38	18	53	66	16	63
Frequency of consumption								
Every day	4	2	10	0	0	0	0	0
Every third day	14	8	21	4	2	3	8	1
Once a week	14	15	15	23	11	26	27	7
Every 15 days	22	37	29	21	50	31	24	13
Once a month	46	38	24	47	37	40	41	62
Occasional	0	0	1	5	0	0	0	17

CDMX: Mexico City; GDL: Guadalajara; OAX: Oaxaca: PUE: Puebla.

**Table 2 foods-10-01284-t002:** Constructs, items and dimensions for the concept of product traditionality evaluated in the survey.

Construct	Item	Reference
Habit	Drinking Mezcal (Tequila) is something that I have been doing for long time.	[11]
	Drinking Mezcal (Tequila) is something that I do without thinking about it.	
	Drinking Mezcal (Tequila) is something that I am used to do since my youth	
Objective knowledge	See Appendix A	Own elaboration
Product involvement	Drinking Mezcal (Tequila) is an important part of my life	[12]
	I just love good Mezcal (Tequila).	
	Drinking Mezcal (Tequila) is a continuous source of joy for me.	
	Decisions about what Mezcal (Tequila) to drink are very important for me.	
	Drinking Mezcal (Tequila) is an important part of my social life.	
Subjective knowledge	I know pretty much about Mezcal (Tequila).	[13]
	I do not feel very knowledgeable about Mezcal (Tequila) (R).	
	I rarely come across a Mezcal (Tequila) that I haven´t heard of.	
	Among my circle of friends, I´m one of the “experts” on Mezcal (Tequila).	
	Compared to most people, I know less about Mezcal (Tequila) (R).	
	When it comes to Mezcal (Tequila), I know a lot.	
Product traditionality *	Mezcal (Tequila) is a beverage historically linked to the region where it is made.	[8]
	Mezcal (Tequila) is a beverage made with a mild technical process.	
	Mezcal (Tequila) is a beverage that has the distinctive sensory characteristics of the region where it is made.	
	Mezcal (Tequila) is a beverage that is present at celebrations, meetings and special dates.	
	Mezcal (Tequila is a beverage that I consume regularly).	
Overall perceived traditionality	I consider Mezcal (Tequila) as a traditional product.	Own elaboration

R = reversed score. * Dimensions not included: Product consumption is something I inherited from my parents; the product is elaborated according to a gastronomic heritage.

**Table 3 foods-10-01284-t003:** Mean values for each construct and traditional dimension per location and product.

	Mezcal Consumers				Tequila Consumers			
Construct	CDMX	GDL	OAX	PUE	CDMX	GDL	OAX	PUE
Overall perceived Traditionality	8.8 ^a^	8.7 ^ab^	8.9 ^a^	8.5 ^b^	8.7 ^a^	8.8 ^a^	8.0 ^b^	8.7 ^a^
Habit	3.4	3.4	3.7	3.3	3.4 ^ab^	3.9 ^a^	3.6 ^ab^	3.1 ^b^
Objective knowledge	8.6 ^b^	8.3 ^b^	11.3 ^a^	8.6 ^b^	10.3 ^b^	11.4 ^a^	10.1 ^b^	7.9 ^c^
Product involvement	4.0 ^ab^	3.8 ^b^	4.4 ^a^	3.8 ^b^	3.3 ^b^	3.9 ^a^	3.8 ^ab^	3.3 ^b^
Subjective knowledge	4.2 ^ab^	3.8 ^b^	4.5 ^a^	4.1 ^ab^	3.5 ^b^	4.2 ^a^	3.5 ^b^	3.6 ^b^
**Dimensions of traditional food product**								
Regular consumption	3.4 ^ab^	3.0 ^b^	3.9 ^a^	3.2 ^ab^	2.1 ^c^	3.4 ^a^	2.5 ^bc^	3.1 ^ab^
Geographic location	6.4 ^ab^	6.1 ^bc^	6.6 ^a^	5.9 ^c^	6.3 ^a^	6.2 ^a^	6.0 ^a^	5.1 ^b^
Low process	4.1 ^b^	4.8 ^a^	5.2 ^a^	4.5 ^ab^	3.5 ^b^	3.0 ^b^	3.6 ^ab^	4.2 ^a^
Sensory	6.1 ^ab^	6.3 ^a^	6.4 ^a^	5.8 ^b^	5.9 ^a^	5.7 ^a^	5.6 ^ab^	5.2 ^b^
YrSpecial dates	4.9	4.7 ^b^	6.3 ^a^	5.0 ^b^	6.3 ^a^	6.2 ^a^	5.8 ^a^	4.9 ^b^

Different lowercase in each row and product indicates significant differences among cities (*p* < 0.05).

**Table 4 foods-10-01284-t004:** Percentage of individuals in each sociodemographic level and consumption habits *, mean values of the different constructs ** and the traditionality dimensions *** within each segment of mezcal consumers.

	Cluster 1 (*n* = 141)	Cluster 2 (*n* = 78)	Cluster 3 (*n* = 181)	Overall Percentage
Sociodemographics * (%)				
City				
CDMX	26.2 ^ab^	38.5 ^a^	18.2 ^b^	25
GDL	25.5 ^ab^	14.1 ^b^	29.3 ^a^	25
OAX	20.6	28.2	27.1	25
PUE	27.7	19.2	25.4	25
Gender				
Men	55.3	65.4	61.9	60.3
Women	44.7	34.6	38.1	39.8
Age				
18–24 years	14.9	9.0	18.8	15.5
25–34 years	41.8	42.3	38.7	40.5
35–44 years	29.1	21.8	30.4	28.3
45–54 years	12.8 ^ab^	24.4 ^a^	6.1 ^b^	12.0
55 or more	1.4	2.6	6.0	3.8
Experience and consumption * (%)				
Experience with the product				
Less than a year	12.8	5.1	13.8	11.8
1 to 3 years	22.7	12.8	22.7	20.8
3 to 5 years	18.4 ^ab^	11.5 ^b^	25.4 ^a^	22.0
5 to 7 years	18.4	20.6	16.6	18.0
More than 7 years	27.7 ^b^	50.0 ^a^	21.5 ^b^	29.3
Frequency of consumption				
Every day	2.8	14.1 ^a^	0.6 ^b^	4.0
Every third day	8.5 ^b^	26.9 ^a^	7.7 ^b^	11.8
Once a week	17.7	21.8	13.8	16.8
Every 15 days	29.9	21.8	27.7	27.3
Once a month	40.4 ^a^	14.1 ^b^	48.1 ^a^	38.8
Occasional	0.7	1.3	2.3	1.5
Construct ** (mean values)				Global mean
Habit	3.3 ^b^	4.7 ^a^	3.1 ^b^	3.7
Objective knowledge	9.0 ^b^	10.6 ^a^	8.8 ^b^	9.5
Product involvement	3.9 ^b^	5.5 ^a^	3.4 ^c^	4.3
Subjective knowledge	3.9 ^b^	5.0 ^a^	3.9 ^b^	4.3
Overall perceived traditionality	8.7	8.7	8.8	8.7
Dimension *** (mean values)				
Regular consumption	3.1 ^b^	6.1 ^a^	2.4 ^c^	3.9
Geographic location	6.4 ^a^	5.8 ^b^	6.2 ^a^	6.1
Low process	3.2 ^c^	4.4 ^b^	5.9 ^a^	4.5
Sensory	6.2	6.1	6.1	6.1
Special dates	5.2	5.4	5.2	5.3

Different lower case within the same row indicates significant differences among clusters. For constructs and dimensions, mean values with different lowercase letters within each row indicate significant differences among clusters.

**Table 5 foods-10-01284-t005:** Percentage of individuals in each sociodemographic level and consumption habits *, and mean values of the different constructs ** and the traditionality dimensions *** within each segment of tequila consumers.

	Cluster 1 (*n* = 167)	Cluster 2 (*n* = 186)	Cluster 3 (*n* = 47)	Overall Percentage
Sociodemographics * (%)				
City				
CDMX	29.3 ^a^	26.3 ^a^	4.3 ^b^	25.0
GDL	14.4 ^b^	35.5 ^a^	21.3 ^ab^	25.0
OAX	30.5 ^b^	10.8 ^c^	61.7 ^a^	25.0
PUE	25.8 ^ab^	27.4 ^a^	12.7 ^b^	25.0
Gender				
Men	49.1	60.2	63.8	56.0
Women	50.9	39.8	36.2	44.0
Age				
18–24 years	28.1 ^a^	15.1 ^b^	10.6 ^b^	20.0
25–34 years	36.5	43.5	38.3	40.0
35–44 years	18.0	23.7	34.0	22.5
45–54 years	10.2	10.2	10.6	10.3
55 or more	7.2	7.5	6.4	7.3
Experience and consumption * (%)				
Experience with the product				
Less than a year	6.0	2.7	6.4	4.5
1 to 3 years	17.4 ^a^	5.4 ^b^	10.6 ^ab^	11.0
3 to 5 years	22.2	13.4	14.9	17.3
5 to 7 years	13.8	19.4	25.5	17.8
More than 7 years	40.7 ^b^	59.1 ^a^	42.6 ^ab^	49.5
Frequency of consumption				
Every third day	3.0	2.7	8.5	3.5
Once a week	10.2 ^b^	20.0 ^b^	36.2 ^a^	17.8
Every 15 days	28.1	31.6	25.5	29.5
Once a month	53.3 ^a^	42.5 ^ab^	25.5 ^b^	45.0
Occasional	5.4	3.2	4.3	4.3
Construct ** (mean values)				Global mean
Habit	3.0 ^c^	3.6 ^b^	4.8 ^a^	3.8
Objective knowledge	9.4 ^b^	10.6 ^a^	8.3 ^c^	9.4
Product involvement	3.2 ^c^	3.6 ^b^	4.7 ^a^	3.8
Subjective knowledge	3.6	3.7	3.8	3.7
Overall perceived traditionality	8.4 ^b^	8.7 ^a^	8.1 ^b^	8.4
Dimension *** (mean values)				
Regular consumption	2.9 ^b^	2.0 ^c^	5.2 ^a^	3.4
Geographic location	6.4 ^a^	5.7 ^b^	4.7 ^c^	5.6
Low process	2.2 ^c^	5.0 ^a^	4.1 ^b^	3.8
Sensory	5.9 ^a^	5.6 ^a^	4.3 ^b^	5.3
Special dates	6.1 ^a^	5.9 ^a^	4.8 ^b^	5.6

Different lower case within the same row indicates significant differences among clusters. For constructs and dimensions mean values, with different lowercase letters within each row indicate significant differences among clusters.

## Data Availability

All data is presented in this manuscript.

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
