# Peer review of "What Turns a Product into a Traditional One?"

_foods, 2021, doi:10.3390/foods10061284_

Round 1

Reviewer 1 Report

My comments are the following:

  1. The aim is not defined in the abstract.
  2. The exact findings are not written in the abstract.
  3. Ethnocentrism should be mentioned in the study, the good example of consumers' perceptions toward traditional and not traditional food products is the following reference and it should be included in the manuscript: Đorđević, Đ., & Buchtova, H. (2017). Factors influencing sushi meal as representative of non-traditional meal: Consumption among Czech consumers. Acta Alimentaria46(1), 76-83.
  4. All interviews were done in person?
  5. Table 4: how clusters 1, 2, 3 are formed? It should be better explained in the manuscript.
  6. The study is interesting and definitely explaining and dealing with issues that are very important at the present time.

Author Response

All the authors of this paper would like to thank the reviewers of this manuscript for their important insights in the improvement of this work.

  1. The aim is not defined in the abstract.

Aim of the study has already been included in lines 16-19   

2. The exact findings are not written in the abstract.

Findings have been added in lines 26-28

3. Ethnocentrism should be mentioned in the study, the good example of consumers' perceptions toward traditional and not traditional food products is the following reference and it should be included in the manuscript: Đorđević, Đ., & Buchtova, H. (2017). Factors influencing sushi meal as representative of non-traditional meal: Consumption among Czech consumers. Acta Alimentaria46(1), 76-83.

Reference has been included in lines 39-41

4. All interviews were done in person?

Explained in lines 123-124

5. Table 4: how clusters 1, 2, 3 are formed? It should be better explained in the manuscript.

Clusters from this section are explained in the methodology section in data analysis, in lines 209-219

6. The study is interesting and definitely explaining and dealing with issues that are very important at the present time.

Reviewer 2 Report

The topic of this manuscript is very interesting and well suited to the Journal but in my opinion it shows some flaws that prompt me to ask the authors a deep work of revision and restructuring of the paper.

My main concern is the following:

According to the authors the aims of the paper is “to examine the impact of the different dimensions of the traditional food provided by Guerrero et al. on the perception of two traditional Mexican alcoholic beverages and to assess the link between these dimensions and the constructs habit, involvement, subjective and objective knowledge”. However when it comes to the results and discussion, it seems that the aims stated in the introduction play only a marginal role while the key result seems to be the cluster analysis for the two products. In addition, I believe that the classification in the three groups does not add anything relevant and does not help to differentiate the two products from the point of view of traditionality nor to characterize them with respect to specific socio-demographic or psychographic characteristics but only according to the level of knowledge and involvement which is not particularly useful from a marketing point of view. Therefore I would urge the authors to deeply restructure the introduction and discussion in order to achieve greater consistency between what is stated as the main aim and the results/discussion.

Others concerns :

The authors should better explain the reasons for the choice of these two particular products. Indeed, the chosen products surely have many typical characteristics of traditional products, but as beverages having a high alcoholic content they also have characteristics which tend to make them different from the definition of food product in a strict sense, moreover because of their nature, as the authors themselves recognize (lines 128-132), two of Guerrero's 7 dimensions cannot be used.

The introduction deserves more attention especially with regard to citations. In fact, several statements are not supported by appropriate and consistent bibliographic references. Starting with the first statement (pg.1 line 1) in which the authors say” Nowadays, there is an increasing demand for traditional food products”, and still on pag. 1 line 44 where it is said “It seems that most consumers associate a traditional food with habit”, I believe it is useful to add a reference to support these assertions.

On pag. 2 lines 88-93 the authors present the 4 cities where the survey took place.  It appears that three of these cities are very involved in the consumption or production of Mezcal and only 1 (Puebla) in the production and/or consumption of Tequila.  Could this choice introduce some bias into the analysis?

I do not understand why the  “Overall perceived traditionality” of the two products was assessed in a nine-point scale and not on a 7 point scale. Please explain

On table 2 in the last item of product traditionality a parenthesis is missing

Line 368: "places place" twice in succession. Please amend

Author Response

According to the authors the aims of the paper is “to examine the impact of the different dimensions of the traditional food provided by Guerrero et al. on the perception of two traditional Mexican alcoholic beverages and to assess the link between these dimensions and the constructs habit, involvement, subjective and objective knowledge”. However when it comes to the results and discussion, it seems that the aims stated in the introduction play only a marginal role while the key result seems to be the cluster analysis for the two products. In addition, I believe that the classification in the three groups does not add anything relevant and does not help to differentiate the two products from the point of view of traditionality nor to characterize them with respect to specific socio-demographic or psychographic characteristics but only according to the level of knowledge and involvement which is not particularly useful from a marketing point of view. Therefore I would urge the authors to deeply restructure the introduction and discussion in order to achieve greater consistency between what is stated as the main aim and the results/discussion.

Aims of the study have been modified in order to better fit with Results and Discussion (lines 109-112)

In the updated version of the manuscript, a more detailed explanation of the relevance of the classification of consumers in segments has been addressed in the introduction section as well as the reason of choosing  knowledge and product involvement as important constructs to be considered in this study.  Lines 50-58, 70-75, 104-108

An additional comment in the Discussion regarding segments has also been placed in lines 459-461, 488-490.

Others concerns:

The authors should better explain the reasons for the choice of these two particular products. Indeed, the chosen products surely have many typical characteristics of traditional products, but as beverages having a high alcoholic content they also have characteristics which tend to make them different from the definition of food product in a strict sense, moreover because of their nature, as the authors themselves recognize (lines 128-132), two of Guerrero's 7 dimensions cannot be used.

A more detailed explanation has been updated in the new version of the manuscript in lines 94-103

The introduction deserves more attention especially with regard to citations. In fact, several statements are not supported by appropriate and consistent bibliographic references. Starting with the first statement (pg.1 line 1) in which the authors say” Nowadays, there is an increasing demand for traditional food products”, and still on pag. 1 line 44 where it is said “It seems that most consumers associate a traditional food with habit”, I believe it is useful to add a reference to support these assertions.

Reference has been included in the new version of the manuscript, line 60

On pag. 2 lines 88-93 the authors present the 4 cities where the survey took place.  It appears that three of these cities are very involved in the consumption or production of Mezcal and only 1 (Puebla) in the production and/or consumption of Tequila.  Could this choice introduce some bias into the analysis?

Puebla is also involved in the production of Mezcal, this statement has been updated in Lines 121-123

I do not understand why the  “Overall perceived traditionality” of the two products was assessed in a nine-point scale and not on a 7 point scale. Please explain

Due to our hypothesis of both Mezcal and Tequila being traditional products, authors decided to use a wider scale in order to have a higher discriminant capacity (wider range in the scale) than the 7 point scale.

On table 2 in the last item of product traditionality a parenthesis is missing

Correction has been made in table 2

Line 368: "places place" twice in succession. Please amend

Correction has been amend line 414

Round 2

Reviewer 1 Report

The manuscript can be accepted.

Reviewer 2 Report

The authors have taken into account the points raised and have arrived at an improved version of the paper which I consider suitable for publication. However, I must note that  the scientific contribution of the manuscript both in terms of theoretical and empirical contribution is in my opinion rather limited.